# Comprehensive Analysis of FBG and Distributed Rayleigh, Brillouin, and Raman Optical Sensor-Based Solutions for Road Infrastructure Monitoring Applications

**DOI:** 10.3390/s25175283

**Published:** 2025-08-25

**Authors:** Ugis Senkans, Nauris Silkans, Sandis Spolitis, Janis Braunfelds

**Affiliations:** 1Institute of Photonics, Electronics and Telecommunications, Riga Technical University, LV-1048 Riga, Latvia; ugis.senkans@rtu.lv (U.S.); nauris.silkans@rtu.lv (N.S.); sandis.spolitis@rtu.lv (S.S.); 2Fiber Optical Sensor Research Group (RTU FiberSens), Riga Technical University, Azenes street 12, LV-1048 Riga, Latvia; 3RTU Communication Technologies Research Center (RTU ComTech), Riga Technical University, Azenes street 12, LV-1048 Riga, Latvia

**Keywords:** road infrastructure monitoring, fiber Bragg grating (FBG), point optical sensors, quasi-distributed optical sensors, distributed optical sensors, Brillouin optical sensors, Rayleigh optical sensors, Raman optical sensors, structural health monitoring

## Abstract

This study focuses on a comprehensive analysis of the common methods for road infrastructure monitoring, as well as the perspective of various fiber-optic sensor (FOS) realization solutions in road monitoring applications. Fiber-optic sensors are a topical technology that ensures multiple advantages such as passive nature, immunity to electromagnetic interference, multiplexing capabilities, high sensitivity, and spatial resolution, as well as remote operation and multiple physical parameter monitoring, hence offering embedment potential within the road pavement structure for needed smart road solutions. The main key factors that affect FOS-based road monitoring scenarios and configurations are analyzed within this review. One such factor is technology used for optical sensing—fiber Bragg grating (FBG), Brillouin, Rayleigh, or Raman-based sensing. A descriptive comparison is made comparing typical sensitivity, spatial resolution, measurement distance, and applications. Technological approaches for monitoring physical parameters, such as strain, temperature, vibration, humidity, and pressure, as a means of assessing road infrastructure integrity and smart application integration, are also evaluated. Another critical aspect concerns spatial positioning, focusing on the point, quasi-distributed, and distributed methodologies. Lastly, the main topical FOS-based application areas are discussed, analyzed, and evaluated.

## 1. Introduction—Common Methods for Road Infrastructure Monitoring

To maintain the safety, quality, and cost-effectiveness of road and transportation infrastructure, it is essential to understand how road infrastructure performs under actual operational conditions. Additionally, regular inspections are performed to assess the quality of pavements after they are deployed and in use by the public [1]. Road infrastructure monitoring at this stage relies on traditional field surveys, such as visual inspection and distress detection by digital imaging technologies, and Falling Weight Deflectometer (FWD) to reveal the structural capacity of the pavement. Ground-penetrating radars (GPRs) and infrared thermography (IRT) are used to assess the condition of sublayers of the pavement. All these methods complement each other and offer an overall understanding of the structural condition of the road pavement [2].

### 1.1. Method 1. Visual Inspection of Road Infrastructure State

Visual inspection of road pavements is among the oldest methods for assessing pavement conditions. This pavement monitoring approach involves the on-site deployment of trained personnel who directly observe the pavement surface to identify visible distress and overall condition indicators [3]. Visual surveys of the pavement can be conducted by walking or driving slowly, while systematically observing and documenting the pavement’s condition indicators [4].

During the visual inspection, the presence of cracks can be identified, and the crack type (longitudinal, transverse, edge, etc.), width (mm), length (mm), and crack density of the overall pavement surface (%) can be determined. Rut, its depth (mm), sagging, and pothole count can be logged. Other aspects of the pavement’s condition, such as surface defects, condition of patches, joints, and other issues, for example, water drainage issues, and others, can also be identified. The information gathered can be quantified using a standardized rating system, such as the Pavement Condition Index (PCI), in which a numeric rating between 0 and 100 (where 100 represents the best possible condition, while 0 represents the worst possible condition) is introduced based on the type, severity, and extent of the damage to the pavement. An example of PCI condition rating categories is shown in Figure 1. The visual inspection method is quite extensively implemented by road infrastructure maintenance agencies worldwide for routine maintenance planning, structural condition monitoring, and quality control [4,5,6].

Despite the method’s cost-effectiveness, rapid execution, and data acquisition, and the direct involvement of humans, which enables swift judgment and possible identification of subtle faults, this method is highly manpower- intensive; furthermore, only surface defects can be easily inspected and identified, while subsurface issues may go unnoticed. Furthermore, this method can be highly subjective, depending on the experience of the inspector, weather, and visibility conditions [3,5,6,8].

### 1.2. Method 2. Surface Condition Assessment with Imaging Technology

The surface condition of the pavement can be assessed with the help of imaging technologies. In the simplest configuration, 2D images can be taken with a digital camera, which could also be mounted onto a surveying vehicle, as shown in Figure 2.

Imaging can also be performed by a laser profiler, which can be employed to accurately measure pavement surface conditions and analyze surface roughness, rutting, and cracks. A laser profiler operates by projecting a light strip on the pavement’s surface, and detectors capture the reflected image. A 3D profile of the pavement is acquired by measuring the distance from every sampled point on the pavement to the detector. The laser profiler is mounted onto the vehicle, and the continuous profile of the pavement is acquired while it moves along the road [9,10].

The acquired data can later be analyzed manually by engineers or with automated systems that are able to analyze acquired images and identify defects on the surface, similarly to regular visual inspections [5,6,11].

Even though, in theory, automated surveying systems would simplify the condition assessment of the pavements, as of this time, there are no available automated pavement condition assessment solutions capable of rating the condition completely autonomously. Furthermore, their ability to detect distress is limited to certain types of faults on the pavement’s surface. Additionally, environmental conditions can significantly affect the data, which could lead to an incorrect assessment of the pavement’s condition [5,12].

### 1.3. Method 3. Deflection Test for the Integrity Assessment

FWD (visible in Figure 3) is a non-destructive testing (NDT) approach that is widely used for the evaluation of the structural condition of road infrastructure [6,8,13,14,15,16,17,18,19,20,21]. An FWD device operates by replicating the moving vehicle’s weight through a controlled pressure, usually in the range of 5 to 150 kN [13,14,15] for the duration of 20–40 ms [13,15]. A known weight (usually 50–700 kg) is released in free-fall motion from a set height onto the circular loading plate that is positioned on a pavement surface, producing a transient load impulse that is similar to a moving vehicle both in the load magnitude and the duration. Different load levels to simulate traffic loads from light to very heavy can be achieved by varying the drop height and weight [13,14,15,16].

The surface deflections caused by the drop of the weight are measured by an array of geophone or seismometer sensors [17] placed at various locations from the loading point (e.g., 0, 200, 300, 450, 600 mm, etc.) [14,16]. The magnitude of the measured deflections usually ranges from 0.2 to 2 mm at the loading center [13,14,15,16]. The load cells and deflection sensors are highly accurate for FWD. For example, the load cell’s accuracy is about +/−2%, while the accuracy of the measured deflections is around 2 µm [14]. Several factors that can affect the measured deflections are pavement thickness, size, and type of pavement, as well as climatic factors such as temperature and seasonal influences [8,16,18].

The structural properties of the pavement layers are indirectly determined by analyzing the acquired deflections. From the deflection data, each layer’s moduli can be back-calculated, and the assessment of the pavement’s structural capacity can be performed. Therefore, implementing the FWD analysis makes it possible to determine the elastic modulus of each pavement layer, determining the load-bearing capacity and stiffness of the pavement, identifying the weakest layer, estimating the remaining life cycle of the pavement, and determining the thickness of the overlay. The conventional understanding is that the load-bearing capacity of the pavement is inversely proportional to the deflection. Furthermore, in good quality, properly built pavement, the deflections should be minimal, and the pavement should exhibit an elastic response to applied load. However, in deteriorated pavements, deflections under applied load are significantly higher, and only a portion of the deformation is recovered; the remaining deformation is retained [8,13,16,19].

The parameters that are used as guidelines for the overall assessment of the structural condition of the pavement layers are derived from the acquired deflection curve. Some of these parameters, which serve as a basis for further assessment of the structural condition of the pavement, are: the Surface Curvature Index (SCI), which indicates the stiffness of the asphalt layer; the Base Damage Index (BDI), which indicates the condition of the lower bearing layers of the pavement; and the Base Curvature Index (BCI), which indicates the stiffness of the base layer [8].

Due to its cost-effectiveness, reliability, and accuracy, the FWD is one of the most universally adopted methods for structural condition monitoring of asphalt pavements and is considered the standard method for deflection testing [3,6,8,11,13,15,16,19]. However, the FWD has some disadvantages, such as the necessity for traffic lane closures, while deflection tests are performed [19,20]. The FWD offers point measurements of the deflection, usually performed at intervals of 25 to 50 m [19]. As a result, some damage may be overlooked between two measurement points. The FWD is highly sensitive to the overall surface smoothness of the pavement. The FWD can be successfully used to survey road infrastructure at certain locations (project-level) and sections, but it is challenging to achieve that in network-level monitoring applications [11,19].

A new type of deflection testing device, which has generated substantial interest and has already found real-world applications, is the Traffic Speed Deflectometer (TSD), which, unlike the FWD, is mounted directly onto a surveying vehicle and can perform deflection tests at traffic speeds (80–100 km/h) [19,20,21]. The TSD applies a continuous load to the road pavement through its wheels and measures the deflection response by using an array of Doppler laser sensors. These sensors are installed in such a way that they can measure deflection velocity at various locations from the TSD’s load axle, similarly to the FWD. The deflection slope can be acquired by measuring the deflection velocity and dividing it by the horizontal speed of the vehicle. From the deflection slope, the deflection basin can be acquired. Even though the TSD acquires similar deflections to the FWD, there is still some debate about the accuracy of the TSD. Additionally, no international standard has been developed for using TSD data for monitoring asphalt pavements, unlike the FWD [22]. Nonetheless, the TSD has already been successfully exploited in various countries. The main advantage of the TSD over the FWD is that it can acquire data at traffic speeds, meaning that it possible to implement this method for the purpose of network-wide monitoring of road infrastructure [19,20,22].

### 1.4. Method 4. Ground-Penetrating Radar (GPR) for Subsurface Condition Monitoring

GPR is a geophysical NDT approach used to monitor the asphalt pavement’s subsurface conditions by utilizing the radar antennas that emit electromagnetic impulses in a range of frequencies between 0.5 to 2 GHz. GPR transmits EM impulses into the pavement and receives the reflected signal, as depicted in Figure 4. The reflections occur at the interfaces between the materials with differing dielectric permittivity. The analysis of the propagation time and amplitude of the reflected signals allows for an estimation of each layer’s thickness, assessment of the material’s properties, and identification of voids or water ingress, depending on the dielectric permittivity of the material. GPR provides continuous profile information on the pavement’s subsurface and is often carried out in tandem with a deflection test to assess the condition of the pavement [23,24].

The penetration depth of GPR is usually around 0.5 to 1 m, which is adequate to cover the asphalt layers and the base layers of the pavement, and the resolution achievable for the layer’s thickness measurement is 2–5 cm. The penetration depth and resolution depend on the frequency of the emitted EM impulses; lower frequencies allow for higher penetration depths but with lower resolution, while higher frequencies offer higher resolution with lower penetration depth [25,26].

GPR has a wide range of applications in the road infrastructure maintenance industry, such as thickness mapping, which enables the identification of layer variations across the pavement. GPR can be utilized to identify defects, such as voids, cracks, moisture concentration, and stripping damage within the layers of the pavement. However, distinguishing between specific details or factors that cause a certain fault in the pavement can be challenging, since different structural anomalies in the pavement structure could elicit a similar response in the GPR signal received. A substantial advantage of GPR is that it can be mounted onto a vehicle, and the data acquisition can be performed at traffic speeds. GPR offers high accuracy for thickness measurements with about 5–10% deviation from the actual thickness of the measured layer, which often corresponds to an error of a few millimeters [25,26,27].

### 1.5. Method 5. Infrared Thermography (IRT) for Temperature Variation Analysis

IRT is an NDT technique used to measure the temperature variations of the pavement’s surface to identify certain underlying defects in the pavement’s structure. IRT can be implemented in two main approaches: passive IRT, which exploits natural daily thermal cycles to increase the temperature of the pavement and then allows it to cool down, and active IRT, which makes use of an external thermal energy source to raise the pavement’s temperature artificially [28,29].

The IRT device captures thermal radiation that is emitted from the road pavement’s surface and produces thermal maps from this information. Usually, areas within a defect, such as a void, moisture packet, debounded or delaminated area, or fracture, will transfer heat at a different rate than an area without any discontinuity, because the defective area disrupts the heat flow in the material, producing thermal contrast on the surface, as depicted in Figure 5. In the figure, red color represents high-temperature regions, while blue indicates low-temperature regions. IRT devices are precise and can measure temperature contrasts of 0.2 to 1.2 °C [30]. Employed in practice, IRT has proven to be quite effective for pavement monitoring [28,29,31].

The main advantages of IRT are that this approach is non-contact, enables rapid data acquisition in real time with minimal or no traffic disruptions, and it can be mounted onto a surveying vehicle or unmanned aerial vehicle. IRT allows for the detection of hidden defects in structure and offers high spatial resolution [28,30].

The disadvantages of IRT, however, are limited penetration depth, which is usually 40–70 mm [28,30], and sensitivity to environmental and weather conditions, such as ambient temperature, wind speed, solar heating, and humidity. Furthermore, it may be challenging to interpret the results because thermal contrast in acquired data may be caused by factors other than structural defects of the pavement. It is also challenging to estimate the severity of any defects just by using IRT, so it is often incorporated with GPR to quantify the extent of certain defects of the pavement’s structure [27,28,30,31].

### 1.6. Traffic Monitoring Methods for Pavement Management Systems

Inductive loop sensor systems are a well-established vehicle detection technology that, over the years, have become the most widely used solution in traffic management systems. Inductive loop sensors can detect the presence of vehicles and count them. Furthermore, in the loop pair configuration, the system can also estimate the speed, length, axle count, and spacing of moving vehicles [32].

An inductive loop is constructed from one or more loops of insulated conductors that are installed into a shallow cut in the pavement, and the cut is resealed afterwards. The loop is energized with a high-frequency alternating current signal that generates an electrical field around the loop. When a vehicle that has metallic parts passes over the section of the sensor, the Eddy currents are induced in the metallic parts, which in turn generate an opposing magnetic field. This causes the net inductance of the loop to decrease in the event that the vehicle is present over the inductive loop. The inductance change causes a shift in the loop’s circuit’s oscillating frequency or amplitude, which is detected by electrical equipment [32,33].

Due to their proven reliability, accuracy, and effectiveness, inductive loop sensors are widely used for traffic and road pavement management systems. They offer advantages such as the ability to work in darkness, glare, perception, and fog, unlike digital imaging, infrared, laser, and radar systems [32,33].

Weigh-in-Motion (WIM) systems are broadly implemented and are a critical part of pavement management systems for providing real-world data on traffic loads, which is the primary aspect of pavement deterioration. Their main purpose is to capture and record the weight of a vehicle at normal traffic speeds without the requirement for it to slow down or stop completely. WIM employs sensors embedded into the pavement to measure the axle loads of vehicles and estimates their weight from the acquired axle load data [34,35,36]. To correctly assign each load event to a specific axle, the WIM system must identify the beginning and end of each event. To do so, WIM systems also incorporate triggering devices, such as inductive loops, laser sensors, infrared sensors, and digital cameras, before and after the WIM sensor, to detect when a vehicle enters and exits WIM and determine the vehicle’s axle count and spacing [32,33,34].

Various sensor technologies are implemented in WIM systems to measure axle loads; the most popular among them are Piezoelectric Polymer Strips, Quartz Piezoelectric Sensors, and Strain Gauge Plates. Furthermore, there are also emerging sensor technologies in this sphere, such as Strain Gauge sensors, Fiber-Optic Sensors, and others [35,36].

The main applications of WIM systems are traffic load estimation and pavement design methods, such as the Mechanistic-Empirical Pavement Guide (MEPDG), which uses data from WIM to predict pavement damage accumulation. Pavement performance monitoring systems exploit the data to analyze how measured traffic loads change over time. For example, if, over a period of time, the average detected loads increase, this could be an indicator of distress, such as cracking and rutting of the pavement, which helps estimate the pavement’s remaining life cycle and plan maintenance. Lastly, WIM systems are used to protect the pavement by enforcing the weight limits on roads [34,35,36,37].

## 2. Fiber-Optic Sensor Technologies in Road Monitoring Applications

One of the more topical choices lately in road infrastructure monitoring is fiber-optic sensing. As optical sensors function by detecting changes in light signal properties, and hence phase, intensity, wavelength, or polarization, it is also possible to estimate the impact of the interaction of physical, biological, or chemical processes [38,39]. Birefringent fiber-optic sensors [40,41,42] offer a solution for monitoring various physical parameters. These sensors exploit the property of birefringence, whereby light polarized in different directions travels at different speeds through certain optical fibers. When the fiber is subjected to strain or other physical influences, the difference between the two refractive indices changes. Studies [40,41,42,43] have shown that parameters such as temperature, strain, pressure, and tensile load can be measured—all of which are relevant for road SHM applications.

Among multiple optical sensor technologies, FBGs, Raman, Rayleigh, and Brillouin optical sensors are among the most commonly used ones in SHM, environmental sensing applications of specific industrial solutions [44,45,46,47]. The operation of each one of these technologies relies on light reflection or light scattering principles to ensure precise point or distributed parameter monitoring. The following subsections are devoted to explaining these optical sensing technologies, highlighting their working principles and application areas.

### 2.1. FBG Fiber-Optic Sensors for Monitoring Applications

FBG-type sensors are one of the topical technologies used in sensing applications, mainly due to their precision, multiplexing capabilities, and utilization potential in challenging environments, such as places with high temperatures or a lack of close-by power supplies [48]. FBG sensors, in essence, are created by inscribing a specific grating-periodic variation structure within the optical fiber that corresponds to the refractive index, thus reflecting certain wavelengths of light while transmitting others. When optical fibers with such FBGs are subjected to external physical changes, for instance, temperature, strain, or vibration fluctuations, the reflected optical signal shifts, ensuring accurate measurements of those parameters that are being monitored [49,50]. Commercial FBG sensors for strain monitoring are shown in Figure 6. FBG Sensor A: Surface mounted (bolted/weldable onto flat surfaces) sensor; B and C: embedded type sensors; D: mini surface (blued/weldable) sensor; E: Patch strain sensor (embeddable within the structure); F: FBG Arrays: Several FBGs on the same fiber (typically for laboratory experiments).

Hence, the Bragg wavelength formula [51] is important in fiber-optic FBG systems to estimate the central wavelength of the light signal that is reflected by the grating inscribed in the fiber core. This allows us to understand the operation and impact of the measured parameter, such as temperature or strain:(1)$$\lambda_{B}\;=\;2n_{eff}\Lambda$$
where *λ_B_* is the Bragg wavelength (reflected wavelength), $n_{eff}$ is the effective refractive index of the fiber core, and Λ is the grating period—the spacing between the refractive index variations.

To better understand the wavelength shift due to the strain and temperature (yet without taking into account interaction with the host material and its properties), it is vital to estimate the response to both strain and temperature; therefore, Equation [51,52] can be used:(2)$$\frac{\Delta \lambda_{B}}{\lambda_{B}}=\left(1-p_{e}\right)\varepsilon +\left(\alpha +\zeta \right)\Delta T$$
where $\Delta \lambda_{B}$ is the shift in Bragg wavelength, $p_{e}$ is the photo-elastic coefficient, ε = the mechanical strain of an FBG used, α is the thermal expansion coefficient of an FBG, ζ is the thermo-optic coefficient, and ΔT = *T* − *T_B_* or temperature change, where *T* is the measured temperature and *T_B_* is the reference or base temperature at the start of the measurements. Hence, the coefficient reference values previously used are $p_{e}$ = 0.22, α = 0.5 × 10^−6^/°C, ζ = 7 × 10^−6^/°C.

As the FBGs in SHM solutions, such as road monitoring, are embedded in the asphalt pavement layers, it is also important to consider the host material effect [52]; thus, the calculations also need to be adjusted.(3)$$\varepsilon_{hm}=\frac{1}{1-p_{e}}\left\{\frac{\Delta \lambda_{B}}{\lambda_{B}}-\left[\left(\alpha +\zeta +(1-p_{e})(\alpha_{h}-\alpha \right)\right]\Delta T\right\}$$
where $\varepsilon_{hm}$ is the mechanical strain expansion coefficient of the host material and $\alpha_{h}$ is the thermal expansion coefficient of the host material.

When carrying out SHM of the structures, such as roads and others, temperature compensation is critical to ensure that the true strain or the strain induced by the subject is measured accurately, without the impact caused by the changes in temperature. Therefore, the use of dual FBGs that are aligned at an angle allows the strain measurement component to be isolated by using Equation [53]:(4)$$\varepsilon L=\;\frac{\frac{\Delta {\lambda_{b}}^{A}}{{\lambda_{b}}^{A}}-\frac{\Delta {\lambda_{b}}^{B}}{{\lambda_{b}}^{B}}}{\left(1-\;p_{e})(1-cos^{2}\theta -\nu_{12}sin^{2}\theta \right)}$$
where *εL* is the true strain with the thermal effect removed and $\Delta {\lambda_{b}}^{A}$ and $\Delta {\lambda_{b}}^{B}$ are the wavelength shifts of FBG 1 and FBG 2, reflecting the peak shift. ${\lambda_{b}}^{A}$ and ${\lambda_{b}}^{B}$ are the central Bragg wavelengths of FBG 1 and FBG 2, or the reference, unshifted wavelengths. $p_{e}$ is used as the optical-fiber photo-elastic coefficient, θ is the angle between FBG-2 and the loading or the FBG-1 that is aligned with the loading direction. $\nu_{12}$ is the Poisson’s ratio of the host material (used material).

The wide range of potential of FBG sensors has led to their vast adoption across various industry sectors. For SHM applications, they are used to assess the integrity of infrastructure such as buildings, bridges, tunnels, and roads by detecting strain and other parameters [54,55]. For instance, Carani et al. [56] undertook an in-depth study of sensors, such as FBG embedment into composite structures for the purpose of structural SHM, proving their capabilities in critical structure safety maintenance.

The energy sector is also interested in the use of FBG for the monitoring of electrical machinery and devices. Suryandi et al. [57] investigated the integration of FBGs in electric machinery, highlighting their potential over conventional mechanical sensing or predecessor sensing technologies, while also enhancing monitoring capabilities. Similarly, FBG sensors have proved useful in geotechnical applications. As shown by Singh et al. [58], this sensor technology is useful for geotechnical health monitoring, including the detection of ground movement and slope stability evaluation, while providing data reliability and an acceptable precision rate.

Regarding road infrastructure, FBG optical sensors have been a particular focus of study by various researchers. For instance, [59] our own researchers previously implemented FBG strain and temperature sensors 25 mm into a cement-treated recycled asphalt pavement (RAP) layer on a national highway in Latvia. FBGs successfully measured vehicle-induced strains and temperature changes in real time, demonstrating durability and precise operation under heavy loads. Kara De Maeijer et al. [60] installed FBGs in three asphalt pavement layers—at both surface and base layers—in Belgium. Hence, different methods were applied to monitor strain progression during asphalt layer construction. Rebelo et al. [61] recently also studied FBGs and their installation, while focusing on the positioning and distance investigation between them. These experiments were conducted on a highway in Portugal; therefore, interest in FBG realization for road SHM applications has been topical throughout Europe.

Lastly, it is important to analyze case studies that report measurement accuracy validated against a reference instrument under a controlled environment. Thus, Liu et al. [62] undertook a comparative case study to validate a strain monitoring solution for asphalt pavement using an FBG strain sensor. In this study, FBG and resistive strain gauges were compared by embedding them into an asphalt specimen and performing a four-point bending test. The acquired strain values from both FBG and resistive strain gauges exhibited a strong correlation. Furthermore, it was noted that FBG possessed better stability with increased specimen temperature and showed stronger agreement with simulated strain values than the resistive sensor. Therefore, FBG would be a better approach for horizontal strain measurements of asphalt pavement. Considering DFOS, Mustafa et al. [63] presented a Rayleigh scattering-based monitoring solution to measure internal strains in asphalt pavements by employing DFOS and foil-type resistive electrical strain gauges. Similar to the case of FBG sensors [62], the experimentally measured strain data showed a strong correlation between DFOS and resistive strain gauges. Additionally, [64] research showed an FBG temperature sensor that was benchmarked against thermocouples on asphalt/concrete specimens, achieving an average error of ±0.13 °C and RMS variation below 1 °C, and Rebelo et. al. [65] demonstrated an FBG strain sensor co-located with an electrical strain gauge in a four-point bending setup, showing a 0–1.3% difference and a standard deviation of 2–3 µε across repeated loading cycles. These examples demonstrate the high accuracy of FBG-based measurements under controlled temperature and mechanical loading conditions and clarify possible minor deviations due to local strain gradients.

### 2.2. Raman Fiber-Optic Sensors for Monitoring Applications

Raman optical sensors, in particular, are useful for temperature monitoring, also known as Raman Distributed Temperature-Sensing (RDTS) solutions. The operation principle is based on the Raman scattering phenomenon, in which optical light interacts with certain molecular vibrations that are observable in optical fiber, therefore resulting in shifted frequency backscattered light. Precise temperature measurements along the fiber are enabled, given the intensity ratio of anti-Stokes to Stokes, whose components are temperature-dependent [44,66,67,68,69].

Like familiar optical sensing technologies, RDTS ensures immunity to electromagnetic interference, thus helping in situations where electronic sensors might fail, not work at all, or lack the necessary power. Such scenarios are also often encountered at remote locations where certain roads are being built. RDTS-based technology has assisted in addressing difficulties related to the spatial resolution and reachable sensing area distance. Monitoring of this type can be challenging for larger distances due to the choice between the signal strength and pulse width. This challenge has recently been addressed by using Raman scattering and multimode optical fiber with an operating range of 800 nm, achieving around ±3 °C precision and 8-m spatial resolution over a few hundred meters [54,55].

When planning RDTS-based sensor systems for infrastructure monitoring, it is important to understand the appropriate calculation, such as the anti-Stokes/Stokes intensity ratio (*R*(*T*)) calculation [70,71,72,73]:(5)$$R(T)\;=\;\frac{I_{aS}}{I_{S}}={\left(\frac{\lambda_{S}}{\lambda_{aS}}\right)}^{4}\exp\left(-\frac{h\Delta \nu }{kT}\right)$$
calculated between the anti-Stokes signal ($I_{aS})$ and the Stokes signal ($I_{S})$. $\lambda_{S}$ is the wavelength of the Stokes Raman backscattered light and $\lambda_{aS}$ is the wavelength of the anti-Stokes Raman backscattered light. h is used for Planck’s constant (6.626 × 10^−34^ joule-hertz^−1^ (or Joule-seconds)), and Δν is the frequency shift between incident and scattered light (Raman shift). k is used to describe Boltzmann’s constant (1.38 × 10^−23^ J/K) and T is the absolute temperature (Kelvin (K)). Another important calculation, when using Raman-based optical sensors for infrastructure, such as road monitoring, is the use of temperature calculation/demodulation T along the fiber [71,73,74]:(6)$$T={\left[\frac{1}{\theta }-\frac{k}{h\Delta \nu }ln(\frac{R\left(1\right)}{R\left(\theta \right)})\right]}^{-1}$$
where θ is the known, fixed reference temperature (in Kelvin), $R\left(1\right)$ is the intensity ratio at the reference point, or known calibration temperature, and $R\left(\theta \right)$ is the intensity ratio at the unknown temperature location.

Lastly, a critical parameter is the spatial resolution, especially for understanding the exact location of the induced parameter impact point, thereby understanding the spacing between adjacent measurement locations. The RDTS spatial resolution depends on whether Raman Optical Frequency Domain Reflectometry (ROFDR) is used or Raman Optical Time Domain Reflectometry (ROTDR). For temperature sensor realization, if long distances (tens of km) are needed for the monitoring, then the ROTDR approach with pulsed lasers would be considered, yet in shorter distances (a few km), amplitude modulation with continuous wave (CW) lasers in ROFDR systems would be considered [74].

For ROTDR-based systems or pulse-based systems, as the principle behind Raman-based sensing is to use backscattered light to estimate the location where the scattering occurs, it can be noted that the distance between two points—the distance between the location in which Raman scattering occurs and the transmitting end point—is *L*. Hence, the time that is needed for the pulsed light detection by the photodetector is *t*. Given that the corresponding light’s pulse has been transmitted a distance of 2 *L*, it is possible to pinpoint the position *L* at which the Raman scattering occurs in an equation [75]:(7)$$L\;=\;\frac{c\cdot t}{2n}$$
in which *n* is the refractive index of the fiber and *c* is the speed of light in a vacuum. This is particularly important for road infrastructure-based solutions, as accurate temperature detection is crucial, and the allocation of temperature shifts is a key calculation.

However, in ROFDR-based Raman sensor systems, if the temperature deviations occur in distances that are less than the spatial resolution, we cannot be certain that the measured temperature is the correct one. For this reason, spatial resolution can then be described as [74]:(8)$$\Delta z=\frac{c}{2n_{gr}}\frac{1}{f_{mod,max}}$$
where $f_{mod,max}$ is the maximum modulation frequency, *c* is the speed of light in a vacuum, and *n_gr_* is the group refractive index of the fiber core.

Additionally, in infrastructure monitoring, RDTS sensors have also been studied in monitoring temperature shifts in dam structures, assisting in the early detection of seepage or structural integrity defects and anomalies [67,76]. However, despite the potential of RDTS sensor systems, there are still some drawbacks, such as a relatively weak Raman backscattered signal that can limit the sensing distances, thus affecting the accuracy. Modern approaches focus on improving signal-enhancing techniques by integrating hybrid sensing methods to overcome those limitations in monitoring applications [44].

In road SHM solutions, Raman-based optical sensors have been used in adjacent road infrastructure, such as tunnel fire detection [71]. RDTS-based fibers in vehicle transport tunnels enable early fire detection as fibers are embedded in the longitudinal direction along the tunnel linings.

Based on the information evaluated in this section, it is clear that there are potential adaptation strategies for RDTS systems in road infrastructure as well. Localized temperature fluctuations (often around 0.1 °C) over distances of up to 10 km can be recorded [44,74,75]. This can be applied to detect freeze-thaw cycles, analyze thermal cracking, or identify unusual temperature changes beneath asphalt layers. Although direct RDTS-enabled sensor systems for monitoring road infrastructure have not yet been widely reported, adapting existing solutions, such as those demonstrated for tunnels or other structures, can facilitate new research to monitor the thermal health of road infrastructure as well.

### 2.3. Rayleigh Fiber-Optic Sensors for Monitoring Applications

Rayleigh-based distributed optical sensors have been studied and used as a solution for SHM in civil engineering applications, such as roadways, tunnels, and bridges [77,78]. The working principle relies on the Rayleigh scattering process, in which light is elastically scattered due to the microscopic variations occurring in the fiber’s refractive index, hence enabling the capability to detect changes in temperature and strain within a certain length of the optical fiber [45].

Spectral shift Δ*ν*_*R*_ measurements are made by the calculations of the Rayleigh backscatter distributed measurement interrogator that depends [79] on the variations in the temperature Δ_T_ and the strain Δ_ε_, hence:Δ*ν_R_* = *C*_ε_Δ_ε_ + *C_T_*Δ_T_(9)
where *C*_ε_ is the strain sensitivity coefficient (typically shown in GHz/(µm/m)) and *C*_*T*_ is the temperature sensitivity coefficient (shown in GHz/°C). Therefore, when performing SHM, strain can be deduced from the spectral shift, and the increment of total strain can be stated as:(10)$$\Delta_{\varepsilon }=\frac{\Delta V_{R}}{C_{\varepsilon }}-\frac{C_{T}}{C_{\varepsilon }}\Delta_{T}$$

One of the main reasons why Rayleigh-based optical sensors are used is their high spatial resolution, which can reach a millimeter-to-centimeter diapason [45,80]. Optical Frequency Domain Reflectometry (OFDR) technology ensures that such fiber-optic sensors are effective in detecting structural shifts, such as strain-induced cracks or variations. High resolution is critical for early damage assessments in concrete structures, where such cracks form [77].

Thus, in OFDR-based Rayleigh-distributed sensor systems, spatial resolution estimation is also important, and can also be described [81,82] as:(11)$$\Delta z=\frac{c}{2n_{g}\;\Delta f}$$
where *c* is the speed of light in a vacuum, $n_{g}$ is the group refractive index of the fiber core, and Δf is the frequency range of the OFDR used.

Similarly, Rayleigh-based optical sensors allow for real-time monitoring capabilities. This can be provided by continuously analyzing the backscattered light, thus gathering immediate feedback on structural changes, assisting in rapid maintenance actions [83]. Therefore, a dynamic environment and objects that are subjected to different kinds of loads, such as roadways and bridges, can benefit from Rayleigh sensor-based capabilities, as traffic loads and environmental factors affect the integrity of the materials and structures [78,84].

Distributed Rayleigh-based optical sensors provide the capability to cover large areas, for instance, a line of road, where there is no need for numerous discrete point sensors. The benefits of this are not only a simplified network structure but also cost savings. The capability to monitor the entire structure with a single optical fiber makes the Rayleigh-based optical sensor an effective solution for large-scale SHM applications [45,84].

To understand the maximum potential sensing range ($L_{max}$) or coherence length, the differential delay for the interferometer can be estimated using Nyquist sampling [85]:(12)$$L_{max\;}=\frac{ct_{g}}{4n_{g}}$$
where *c* is the speed of light in a vacuum and $t_{g}$ and is the differential delay introduced by the interferometer.

However, despite the advantages, there are also challenges associated with the use of Rayleigh-based sensing. For instance, the interpretation of backscattered signals can be challenging and complex, as optical signal processing algorithms need to be accurately configured to detect the strain and temperature information. Adding to that, the sensing range of Rayleigh-based sensor systems is typically limited to a few tens or hundreds of meters, or just a few kilometers [86].

A recent study [87] highlighted the potential of Rayleigh-based distributed sensing for pavement engineering to ensure real-time road condition monitoring in long-distance measurements. Another relevant recent study [88] also approved the use of Rayleigh-based sensors via rapid OFDR technology realization, to capture dynamic strain data when vehicle-induced strain responses could be recorded under realistic loading conditions, therefore proving a successful live road traffic scenario realization. Herbers et al. [78] recently undertook research on crack monitoring in concrete structures, localizing micro-cracks (as narrow as 0.2 mm). Such an investigation is also topical for road SHM and the detection of structural damage in its pavement layers. Previously, Chapeleau et al. [89] demonstrated the effective embedment of Rayleigh-distributed optical sensors within asphalt pavement, in the bituminous layers, while also performing fatigue testing.

Rayleigh-based distributed optical fiber sensors ensure a high-resolution, real-time, and relatively cost-effective solution for monitoring the structural health of various structures, such as road infrastructure and others. The ability to detect early signs of damage or the structure’s deterioration provides comprehensive coverage, leading to smart infrastructure development and utilization in safe and long-lasting civil infrastructure.

### 2.4. Brillouin Fiber-Optic Sensors for Monitoring Applications

Brillouin-based distributed optical fiber sensors that utilize Brillouin Optical Time-Domain Analysis (BOTDA) and Brillouin Optical Time-Domain Reflectometry (BOTDR) leverage the stimulated or spontaneous Brillouin scattering effect to measure physical parameters such as strain and temperature over long distances, typically, tens of kilometers with tens of centimeter to meter-level spatial resolution. The capability to simultaneously ensure strain and temperature sensing for the structure makes the technology particularly suitable in monitoring large-scale infrastructures, like roads, pipelines, tunnels, and bridges [90,91,92,93,94,95].

Despite the advantages of BOTDA/BOTDR-based sensing solutions, which offer an extensive range of sensing measurement distances and dual-parameter monitoring capabilities, there are still some limitations in spatial resolution and interrogation speed when compared, for instance, to Rayleigh-sensor-based systems. Thus, hybrid architectures in which Brillouin and Rayleigh combinations are used together can offer advanced signal processing, making such approaches topical also for comprehensive road infrastructure monitoring solutions [91,92,93,94,95,96].

In Brillouin-based sensing, the frequency shift $\nu_{B}$ between the pump and backscattered Stokes light can be described as follows [97,98]:(13)$$\nu_{B}=\frac{2n_{eff}V_{a}\;}{\lambda }$$
where $n_{eff}$ is the effective refractive index of the fiber, $V_{a}$ is the acoustic (sound) velocity in the fiber, and λ is the optical pump wavelength or wavelength of the incident light. For instance, regarding the Brillouin shift sensitivity to strain and temperature, the measured frequency shift depends linearly [97] on both physical parameters—strain (ε) and temperature (*T*). Thus, the equation explaining this process can be described as follows:(14)$$\nu_{B}-\nu_{0}=\alpha_{\varepsilon }\Delta \varepsilon +\alpha_{T}\Delta T$$
where $\nu_{0}$ is the reference Brillouin shift or baseline state (no strain) and ambient temperature, while $\alpha_{\varepsilon }$ and $\alpha_{T}$ are coefficients linking the shift to strain. *ΔT* is used to describe the temperature change when comparing the ambient temperature and the measured one. Δε similarity describes the strain change. Thus, simultaneous monitoring capability of both parameters is also possible. Examples of Brillouin and Rayleigh-distributed sensors for strain (**A**–**E**) and crack (**A**–**D**) monitoring solutions are shown in Figure 7. These sensors are typically used for concrete (**A**–**E**), asphalt (**B**), steel (**A**–**E**), soil (**A**–**B**), timber (**D**), and composite (**A**–**D**) material and construction monitoring [99].

For accurate strain readings, it is also important to distinguish true strain when temperature impact has been compensated. Thus, to extract the independent strain, compensating algorithms and a temperature-compensating reference fiber can be used. Then, a strain term can be isolated, and an equation [97] describing this process could be:(15)$$\Delta \varepsilon =\frac{\nu_{B1}-\nu_{01}-\alpha_{T}\Delta T_{2}}{\alpha_{\varepsilon }}$$
in which $\nu_{B1}$ and $\nu_{01}$ are measured and reference Brillouin frequency shifts in the optical sensor fiber. Hence, $\Delta T_{2}$, the temperature change, is measured in the strain-free reference fiber. To ensure precise spatial localization along the optical fiber, it is also critical to distinguish the position (*d*) along the fiber [97,100] that corresponds to the Brillouin event (for instance, induced strain event), which is calculated using the following formula:(16)$$d=\frac{c\Delta t}{2n}$$
where c is the speed of light (vacuum), Δt is used for the round-trip time delay of backscattered light, and n is the refractive index. Lastly, in the context of SHM (such as road pavement monitoring), by using BOTDA-based sensor systems, it can also be critical to understand the sensing resolution and limitations [97]. Therefore, the spatial resolution (*Δz*) of such a BOTDA system is governed by the pulse width *τ*, hence:(17)$$\Delta z=\frac{c\tau }{2n}$$

One of the practical examples of BOTDA sensor systems that has been studied is the field of concrete structures, as shown in research conducted by Wei et al. [101]. Brillouin-based sensors have been used to detect strain variations caused by corrosion expansion, while achieving a spatial resolution of approximately 1 m, which is sufficient for progressive deterioration tracking. However, despite this, limitations in spatial resolution could necessitate more precise and higher-resolution methods.

Imai et al. [102] demonstrated that it is possible to use BOTDA for asphalt structure deformation monitoring, where a controlled experiment was carried out and embedded BOTDA sensors were used to measure strain and deformation under load. Although the asphalt structure has a variable modulus, the Brillouin-based sensors provided reliable and repeatable strain data readings, validating their robustness for asphalt pavement structure monitoring. Another study, conducted by Liu et al. [103], supplemented the use of Brillouin-based fiber-optic sensors by installing more than 200 m of Brillouin fiber-optic sensors in a subgrade trench. Such a solution allowed for the detection of freezing-thawing cycles and potential cracks, thus showing distributed strain mapping across the pavement and subgrade layers. Lastly, Ou et al. [104] demonstrated that the Brillouin scattering spectrum (BSS)-based technique can be used to detect cracks (<0.1 mm) in concrete overlays on the asphalt. Spectral variation analysis can also be used to identify and quantify crack development, even with 0.75 m spatial resolution conditions.

Brillouin-based optical sensors can provide a long-range solution for SHM in transportation—road infrastructure. When used together with other technologies and methods, such as Rayleigh and/or FBGs, a complete, comprehensive approach that balances spatial coverage and long-distance monitoring can be achieved.

### 2.5. Fiber-Optic Sensor Technology Comparison

As discussed in the previous subsections, it is clear that multiple technologies can be adjusted and used for road infrastructure monitoring purposes. Each has unique advantages that can be more suitable in specific configurations and monitoring needs. Therefore, for a clear view of the main differences, we have gathered and evaluated information regarding the optical sensor technology type such as FBG [51,87,105,106,107,108,109,110,111,112,113,114,115,116,117,118], Rayleigh [86,119,120,121,122,123,124,125,126,127], Brillouin [46,102,103,104,128,129,130,131,132,133,134,135,136,137,138,139] and Raman Scattering [44,67,71,74,75,76] and their correlation to the typical sensitivity in key physical parameters such as strain, temperature, vibration, pressure, and humidity (see Table 1).

From the data collected, analyzed, and shown in Table 1, it is clear that every optical sensor-based technology that might be applied for road infrastructure monitoring purposes has its advantages and main target areas in which specific technology would be most efficiently used.

For instance, FBG-based optical sensors can ensure the optimal operation of vibration, pressure, humidity, strain, and temperature parameter monitoring capabilities. This technology in road applications could be used when only a few measurement points and data recordings from them are needed, for example, when the particular road is not categorized as high traffic and/or an important road, and the estimated expected physical parameter’s negative impact is relatively small. Additionally, for security and observation purposes, such single-point optical sensors could be sufficient, especially on narrow roads and bottleneck-type locations. However, even if multiple FBGs are needed for longer distances, it is still possible to apply certain multiplexing methods, such as wavelength-division-multiplexing (WDM) [140,141], to consolidate them within one single optical data transmission line—transmission medium. This allows for the frequency of resource optimization and improved management of the overall sensor system, as every reflected optical signal of each FBG sensor can be detected, processed, and analyzed at the same time and in the same system, in real time [142].

In distributed fiber-optic sensor systems, the backscatter signal and its characteristics are important when choosing the corresponding technology as a basis for the sensors’ network (see the schematic representation of Rayleigh, Brillouin, and Raman backscattering intensity in Figure 8). In Brillouin-based systems, the ~0.2 m–3 m spatial resolution limit arises from the Brillouin phonon lifetime, creating a trade-off between spatial resolution, signal-to-noise ratio, and frequency estimation precision over long operation ranges. By contrast, Rayleigh-based OFDR relies on coherent Rayleigh pattern correlation with ultra-narrow linewidth lasers to achieve millimeter-scale resolution, although at the cost of a reduced sensing range and longer data acquisition times.

Rayleigh scattering provides the strongest backscattering signal and is elastic, which means that the scattered light retains almost the same energy as the incident photons’ energy, making it suitable for strain and vibration detection while using coherent techniques such as OFDR or phase-sensitive OTDR.

However, in Brillouin scattering, which is inelastic and interacts with acoustic photons, both Stokes (loss of energy and lower frequency) and anti-Stokes (gain of energy and higher frequency) components can be observed in an offset of around 10–11 GHz (0.05–0.08 nm at 1550 nm) [71,74,144,145]. In Figure 9, the experimentally measured spectrum of Brillouin scattering with base signal and anti-Stokes and Stokes components can be seen.

Brillouin backscatter signals are typically weaker than Rayleigh ones by about ~16–20 dB [146,147]. As for Raman scattering, higher-energy phonons are involved, producing Stokes and anti-Stokes that are shifted strongly from the pump (~13 THz frequency shift or ~100 nm wavelength shift at 1500 nm). Their Stokes intensity far exceeds anti-Stokes due to Boltzmann population distributions. Yet the overall intensity of the Raman is the weakest (~20 dB (Stokes), 30 dB (anti-Stokes) lower than Rayleigh. In this situation, sensor systems rely on the Stokes/anti-Stokes intensity ratio for accurate distributed temperature-sensing capabilities [71,74,144,145,146,147].

Comparative costs for sensing systems and sensing points for traditional and fiber-optic road monitoring technologies can be compared in Table 2.

As can be seen in Table 2, the largest costs for both traditional and fiber-optic sensor systems are the costs of the interrogator and controllers. The most expensive interrogator costs are observed for optical distributed scattering sensor technologies (>€100,000), followed by Falling Weight Deflectometer (>€50,000) and weigh-in-motion technologies (>€20,000). On the contrary, the cheapest sensor costs are for distributed sensing, costing up to a few euros per meter or sensing point. Of the traditional methods, the lowest cost per sensor point is Manual inspection, Imaging (digital camera, laser profiler), and Falling Weight Deflectometer, but all of these methods only provide periodic inspection, which means that they must be repeated relatively often.

Considering the costs of the technologies and their advantages and disadvantages, we see high potential for fiber-optic sensor technologies to complement the traditional methods used so far.

Obtain data-based information on the condition and service life of the road surface (including inside the pavement, not just from the top layer)Fiber-optic distributed technologies provide the ability to perform real-time SHM of road pavement along the entire fiber length (from a few meters to 50 km and 100 km)Fiber-optic distributed sensing makes dense sensing (from mm to m scale, thousands of points).

## 3. Fiber-Optic Sensor Placement Within Road Infrastructure Monitoring Solutions

When performing infrastructure monitoring, such as road infrastructure and other types, the strategy of spatial positioning of the fiber-optic sensors, as well as the number used, is important. Commonly, three types of positionings are used—single-point sensors, quasi-distributed and fully distributed fiber-optic sensors, and spatial positioning. Each configuration offers distinct advantages in terms of spatial coverage, resolution, and installation complexity, determining how physical quantities like strain, displacement, temperature, and damage are detected and localized.

### 3.1. Single-Point Fiber-Optic Sensors

Single-point optical sensors, such as individual FBGs, are embedded at specific critical locations, such as certain locations in road, bridge, and tunnel infrastructure that are the most critical or locations that might encounter the highest physical parameter, such as strain, impacts. The conceptual design of the point optical sensor implementation can be seen in Figure 10.

As demonstrated in our previous research [73,156,157], such discrete sensors offer high accuracy (~0.8–36 με) per axle based on the specific vehicle, yet they provide no spatial context beyond their location, requiring careful placement in areas of known structural significance. Therefore, if longer paths of road infrastructure need to be monitored, a hybrid approach with distributed fiber-optic sensors, or distributed only, might be chosen.

### 3.2. Quasi-Distributed Fiber-Optic Sensors

Quasi-distributed fiber-optic sensor systems use multiple fiber segments or FBG arrays or localized distributed (for instance, Brillouin) sensors distributed at intervals, offering mid-range spatial insight while balancing complexity. The concept of a quasi-distributed fiber-optic sensing solution is shown in Figure 11.

One example of such an approach can be observed in the ASPARi project in the Netherlands, where FBGs spaced 10 cm apart within a prefabricated bicycle pavement section yielded temperature and strain profiles through layered pavement monitoring [158]. Similarly, Wang et al. [159] applied BOTDA in an 8 m cement testbed using armoring-wire encapsulated quasi-distributed sensors, each functioning as an independent strain gauge to pinpoint cracks with high local precision.

### 3.3. Distributed Fiber-Optic Sensors

Distributed fiber-optic sensor architectures use techniques like OFDR, OTDR, BOTDA, BOTDR, phase-sensitive OTDR, Raman scattering, or others to offer continuous sensing along the entire fiber, providing full-length spatial profiles [78,89]. The concept of a distributed fiber-optic sensing solution is shown in Figure 12.

For road pavements, a distributed fiber-optic sensor system employing Rayleigh-based scattering [89] was used to detect crack initiation in asphalt subjected to 728,000 heavy vehicle loads. The distributed fibers, embedded near the bottom of the asphalt layer (<1 cm resolution), detected strain changes well before visual cracks became apparent.

### 3.4. Comparisons Between Single-Point, Quasi-Distributed, and Distributed Fiber-Optic Sensor Realization

The three main fiber-optic sensing architectures—single-point, quasi-distributed, and distributed—differ primarily in their spatial coverage, resolution, scalability, and installation complexity. Single-point sensors, such as individual FBGs, are limited to localized measurements and are typically used for monitoring specific high-risk points within the road infrastructure. They offer high accuracy but require precise pre-identification of critical zones.

Quasi-distributed systems expand on this by using multiple discrete sensors (typically FBGs) arranged along the fiber at set intervals. This allows for partial spatial coverage across a road structure, offering a compromise between full coverage of the whole infrastructure (road span, highway, etc.) and cost. They are well-suited for medium-scale monitoring where certain zones, like layers or segments, require repeated sensing along the road length or road depth (layers).

In contrast, distributed optical sensors (based on Rayleigh, Brillouin, or Raman scattering) provide continuous sensing along the entire fiber, enabling measurement of parameters such as strain, temperature, and others with spatial resolutions ranging from meters to millimeters (for more details, see the corresponding section of each technology in this study). These systems are ideal for long-span, linear infrastructure, such as roadways, tunnels, or bridges, where wide-area detection and early warning capabilities are essential. However, they also involve higher data rates, more complex interrogation units, and potentially greater installation effort from road infrastructure managers.

### 3.5. Calibration and Long-Term Drift for Embedded Fiber-Optic Sensors in Asphalt Concrete

Embedded fiber-optic sensors, particularly FBGs in asphalt concrete, require careful calibration and mitigation strategies for long-term drift and environmental cross-sensitivities to ensure reliable measurements. Field implementations, such as the FBG system installed in the previous projects [52,158], have demonstrated the need for temperature compensation or recalibration, as baseline signal drift can occur over time due to temperature fluctuations and the viscoelastic behavior of asphalt [43,158].

Recent advances in temperature-sensing materials, such as iron-ceramic enhanced FBG sensors [160], offer improved high-temperature stability and encapsulation robustness, which could reduce drift in asphalt environments by improving thermal response analysis. Similarly, multiwavelength cascaded FBG arrays inscribed by femtosecond laser and interrogated via swept laser demodulation [161] provide highly precise temperature sensing (more than 8.9  pm/°C with a linearity of 0.989), enabling more effective strain-temperature decoupling in embedded applications.

## 4. Discussion—Perspective Optical Sensor Solutions in Road Infrastructure Monitoring Applications

Regarding the discussed key aspects of fiber-optic sensors and their configurations, there are several topical road monitoring-based applications in which such optical sensors could be used and have been mentioned in this article: namely, traffic volume monitoring, vehicle speed detection, traffic density estimation, and weigh-in-motion for vehicle identification.

### 4.1. Traffic Volume Monitoring Using Fiber-Optic Sensors

Based on the optical sensor technology types discussed in this article and acknowledging their capabilities and advantages, one prospective future research area in road monitoring applications is traffic volume monitoring.

Distributed optical sensors, as well as point optical sensors such as FBG-based ones, can be used as part of the process to measure the number of vehicles passing a specific physical point of interest in the road infrastructure or in a given period (for instance, vehicles per hour or per day: see Figure 13). So far, generally, only the most used roads have been monitored continuously, and such solutions require mechanical and electrical (induction loop) sensors [162,163,164], radars [165,166], real-time recording cameras, computer vision, artificial intelligence solutions, or a combination [167,168,169], therefore also requiring an active, redundant, and consistent power supply. Thus, such monitoring capabilities are not feasible for large-scale road infrastructure monitoring solutions, especially on less observed and active roads.

Optical sensor-based solutions can ensure efficient and targeted solutions for analyzing traffic flow patterns, assist road infrastructure managers and planners in gathering information on peak hour estimation, and understand the changing indicators of road usage trends that might vary according to the seasons, holidays, weather conditions, and other changing aspects. Such a case study [170] has been presented by Luna OptaSense, which deployed a traffic monitoring solution on I-29 Highway, Fargo, North Dakota, USA.

### 4.2. Vehicle-Induced Strain Detection and Movement Speed Calculation with Fiber-Optic Sensor-Based Solutions

Another important aspect is the gathering of physical parameters (such as strain) by using fiber-optic sensors embedded within the road infrastructure. Such strain data can then be used for multiple potential solutions—directed toward the safety, longevity, and structural integrity monitoring of the road infrastructure, materials used, and constructions realized, and, secondly, regarding vehicle and transportation-based data acquisition solutions. For instance, the concept of vehicle-induced strain detection and data correlation to the vehicle movement speed calculation can be observed in Figure 14.

Optical sensors, such as FBG ones, can be used not only for the structural integrity monitoring of the road infrastructure but also for the estimation of vehicle movement speed. As the embedment point location and distance between two separate FBG optical sensors are known (D1 (m) and D2 (m) from Figure 14), once the integration process has been performed, it is also possible to capture data relating to the vehicle movement speeds. For instance, when the vehicle passes the first FBG 1, all the axles introduce a certain amount of microstrain in a certain time slot. The same process is recorded after the vehicle passes FBG 2, yet the time slot has been increased forward. The difference is the time that has passed (T2 (s) − T1 (s)).

Therefore, the vehicle speed (m/s) can also be calculated. A similar estimation can also be made by using only a single-point optical sensor. Then, the difference is calculated not between two separate FBGs, but between two axles of the particular vehicle. In this scenario, resources can be saved by using fewer FBG optical sensors and the available optical frequency spectrum needed; however, the risks of increased secondary strain output are higher, due to the time needed for the normalization of strain-induced oscillations and their detection.

### 4.3. Traffic Density Monitoring with Fiber-Optic Sensors

Similar to traffic volume monitoring, smart optical sensor solutions can also perform traffic density monitoring. To describe and analyze traffic flow theory, aspects such as traffic density can be measured and evaluated to collect data on the number of vehicles for a certain amount of roadway length. While traffic volume monitoring mainly focuses on counting the vehicles that pass a certain point of interest within the road infrastructure, traffic density measurements are made to analyze how vehicles are distributed across the whole measurable road area within a certain period. For this purpose, distributed optical sensors utilizing Brillouin and Rayleigh scattering could potentially be used due to their abilities to ensure continuous strain- and vibration-based spatial profiles along the distance of interest (see Figure 15).

For instance, by embedding such a type of fiber-optic sensor along the pavement structure, the presence of vehicles and their clusters can be detected over several-kilometer-long distances, thus enabling real-time continuous traffic density monitoring. Such sensors can register localized vibration, temperature, or strain data and fluctuations of those physical parameters caused by vehicle-induced impact on the road surface. These data can then later be correlated with the vehicle amount and spacing, and the overall strain distribution on the different layers of the pavement structure. When compared to point-based optical sensors, in this scenario, such distributed optical sensors provide a more precise understanding of traffic jams and congestion patterns, as well as assisting smart road designers in estimating platooning principles within the area.

### 4.4. Weigh-in-Motion with Fiber-Optic Sensors for Vehicle Identification and Big Data Processing

Weigh-in-motion (WIM) applications are aimed at continuous measurements of loads induced by vehicle axles and their own weight, while not negatively affecting traffic flow or stopping certain vehicles, such as trucks, for those inspections. These solutions are necessary to ensure data gathering is used for freight analysis, road infrastructure maintenance planning, as well as control of legal load limits. Distributed optical sensors and single-point optical sensors such as FBGs can be used for this operation, given their durability, precision, and passive nature. They can be installed within the road pavements to assist in the WIM applications for critical points, such as roads and bridges, with set load limits. Strain-sensitive optical sensors respond to vehicle-induced strain values that can be correlated to the vehicle movement speed and recorded strain levels, therefore comparing such data to corresponding calibrated values. The concept of such a solution is shown in Figure 16.

Optical-based WIM solutions can also enable advanced vehicle identification data estimation, gathering, and analysis, correlating weight profiles and signatures with other volatile vehicle parameters such as movement speed, axle spacing, and count, and suspension response data. Investigation of big data frameworks and machine learning algorithms, and their coupling with such optical sensor-based WIM solutions, can assist in the automation of vehicle type categorization, as well as detecting overloaded vehicles and, therefore, violations of laws and security procedures. Additionally, such available data could also be used in transport modeling, planning of maintenance and its predictive schedule graphs, as well as traffic policy and control procedures. All these capabilities would also allow for the modernization of road infrastructure monitoring through the utilization of smart and modern sensor network-based solutions.

### 4.5. Structural Health Monitoring of Road Infrastructure with Fiber-Optic Sensors

For fiber-optic sensor technologies to be adopted on a larger network-level scale in structural health monitoring of road infrastructure, certain quantitative performance benchmarks must be met. Nevertheless, it is important to note that various scenarios, as well as traffic intensity, road quality, quality of the materials used, and environmental aspects, also play a part in the quantifiable criteria, when such an adoption is planned. However, some of these parameters could be stated as reasonable criteria on a larger scale and in the aforementioned conditions. Currently, from a commercial point of view, in road monitoring applications, there are single-point sensors available (with a measurement area of 10–30 cm); thus, a sensor network-based solution that covers the whole driving lane is needed in the near future. Another aspect that is required is a calibrated solution that is simple to configure and use to measure both temperature and strain simultaneously, while distinguishing their individual impacts (compensation mechanism). Current methods rely on multiple sensor utilization, manual calibration, and manual tests.

Point-sensing systems, such as FBG arrays and their interrogation units in a future network-level SHM system, should target ≤1–5 µɛ strain accuracy over >10–50 km fiber length without the need for intermediate amplification. As for the acquisition rates, a sampling rate of 100–500 samples per second should be aimed for every point sensor (based on the specific road type—local, national, highway, etc.). Achieving this will further require advances in ultra-low-noise interrogators, temperature-strain decoupling (compensation mechanism), and power-efficient amplification schemes (if necessary). Such FBG-based sensor systems can be used for smart solutions, such as those described in Section 4.1, Section 4.2, Section 4.3 and Section 4.4 as well as for road SHM.

Regarding distributed sensor systems that utilize Raman, Rayleigh, or Brillouin-based solutions, there should also be certain target markers for future network-level SHM systems. In small sites, such as urban road sections, intersections, and others, high-resolution Rayleigh OFDR systems should be the main solution when sensing distances are 2 km. Hence, mm- to cm- scale strain spatial resolution and ~1 µε strain accuracy should be the target zone. For medium to long distances, such as highway sections, airport runways, etc., Brillouin-based systems should be chosen in BOTDA/BOTDR configurations, where the sensing distance should ensure at least 20–30 km distances with spatial resolution not lower than 1 m and accuracy within the range of 2–30 µε (based on the distance and possibility of additional laser introduction in the system). Thus, if larger distances of sensing networks need to be covered, to ensure the necessary quality parameters, a zone segmentation approach of 20–50 km sectors that are linked to the central data collection and processing could be utilized. Given this, distributed fiber-optic sensor systems are particularly optimal and suitable for road SHM when ensuring large distance monitoring is critical.

While laboratory demonstrations have met or exceeded many of these specifications and criteria individually, achieving multiple ones simultaneously in a real-life field deployment, for an embedded road sensor network, remains an ongoing challenge.

## 5. Conclusions

This study explored the most commonly used methods for structural condition monitoring of road infrastructure, such as visual inspections, the digital imaging approach, Deflection tests, Ground-Penetrating Radar (GPR), and Infrared Thermography Imaging (IRT). A description and the main applications of each method were also presented. Each of the explored methods offers certain advantages and disadvantages, is tailored to specific monitoring purposes and requirements, and has proven to be an effective tool for assessing the condition of road pavements. However, no one method is sufficient to yield a complete understanding of the structural condition of the road pavement; they complement each other, and for effective monitoring of road infrastructure, the different methods usually need to be deployed in conjunction with one another. Though highly effective, these methods do not offer real-time, long-term structural monitoring, and even though some automation is available in terms of data acquisition and analysis, human input is necessary to effectively and reliably detect subtle faults and judge the structural condition of the pavement.

This study also presented a comprehensive evaluation of fiber-optic sensing technologies, namely, Fiber Bragg Grating (FBG), Rayleigh, Brillouin, and Raman-based sensor systems, in the context of road infrastructure structural health monitoring (SHM). Each sensing category was analyzed in terms of its operational principles, relevant physical phenomena, and practical deployment architectures (single-point, quasi-distributed, and distributed). Special attention was paid to the spatial resolution, typical sensing sensitivities, backscatter signal characteristics, and their implications for monitoring key parameters such as temperature, strain, displacement, tilt, vibration, and humidity within road pavement and substructure layers.

The analysis showed that FBG sensors, with their high precision and multiplexing capability, remain highly effective for localized or quasi-distributed sensing scenarios, especially when integrated at critical structural points. Conversely, distributed optical sensors—utilizing backscatter phenomena from Rayleigh, Brillouin, or Raman scattering—provide full-length monitoring along the fiber with variable spatial resolution, making them particularly suited for long-span infrastructure such as roads, tunnels, and bridges.

Additionally, a comparative review of fiber-optic sensor placement strategies highlighted the differences between accuracy, spatial coverage, and system complexity. Real-world deployment examples and peer-reviewed references validated the practical use of these technologies in road infrastructure applications, though certain gaps, such as limited availability of detailed performance metrics in some studies, remain.

This study exhibits that selecting the most appropriate fiber-optic sensing system for road infrastructure monitoring requires a balanced evaluation of monitoring objectives, spatial resolution needs, infrastructure scale, and installation constraints. With the ongoing advancements of optical sensing technologies and interrogation techniques, their integration into modern road SHM systems is expected to grow, enabling more intelligent, responsive, and data-driven infrastructure maintenance frameworks.

## Figures and Tables

**Figure 1 sensors-25-05283-f001:**
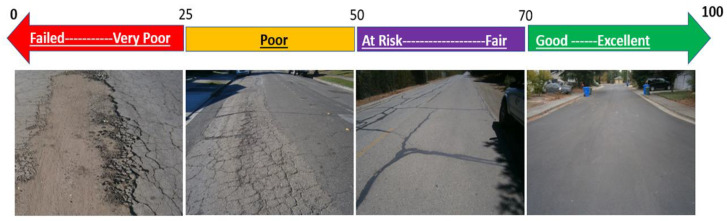
Example of PCI condition rating categories [7].

**Figure 2 sensors-25-05283-f002:**
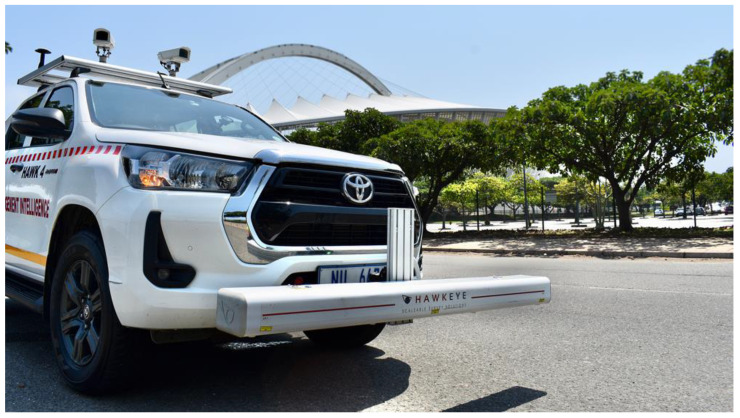
Surveying vehicle with mounted digital cameras and laser profiler [9].

**Figure 3 sensors-25-05283-f003:**
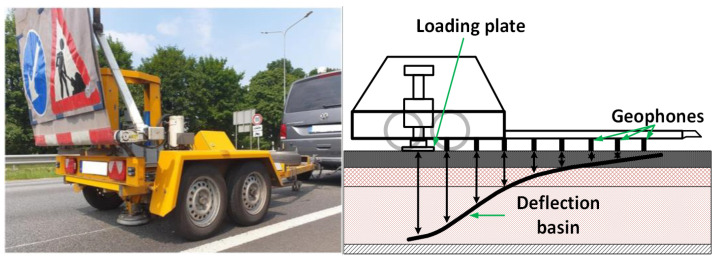
FWD trailer and operational architecture scheme.

**Figure 4 sensors-25-05283-f004:**
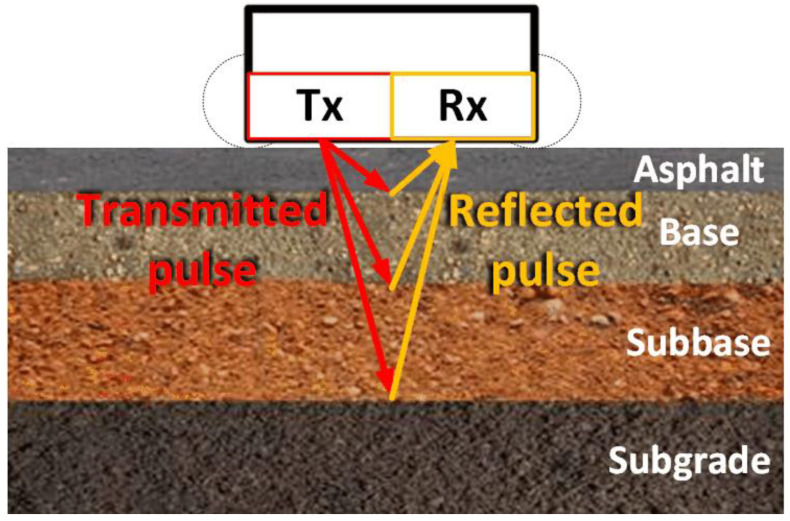
Schematic of GPR.

**Figure 5 sensors-25-05283-f005:**
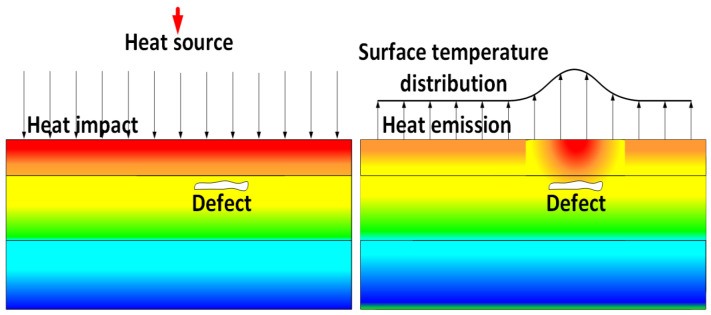
Principle of IRT detection.

**Figure 6 sensors-25-05283-f006:**
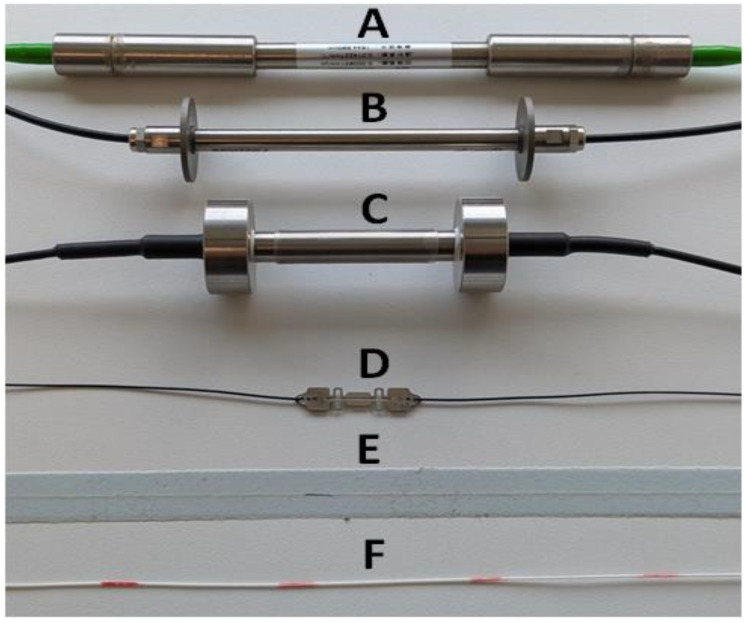
Optical FBG sensors for strain monitoring.

**Figure 7 sensors-25-05283-f007:**
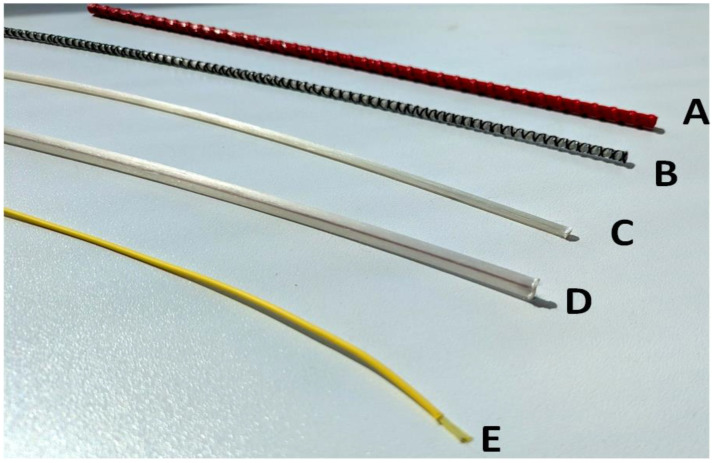
Brillouin and Rayleigh-distributed sensors for strain and crack monitoring.

**Figure 8 sensors-25-05283-f008:**
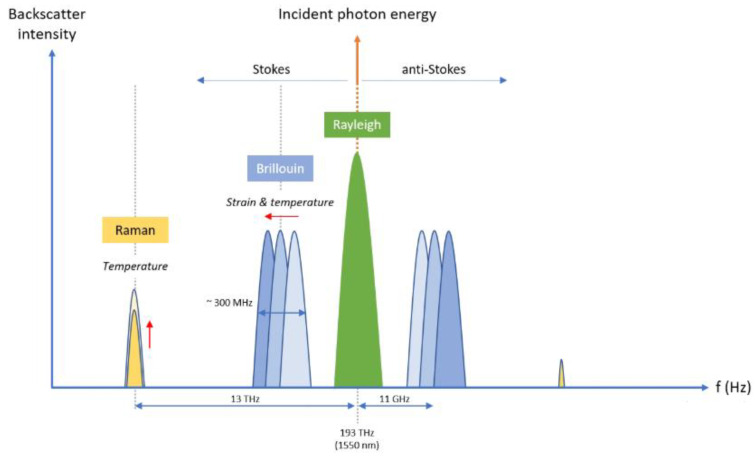
Schematic representation of Rayleigh, Brillouin, and Raman backscattering intensity [143].

**Figure 9 sensors-25-05283-f009:**
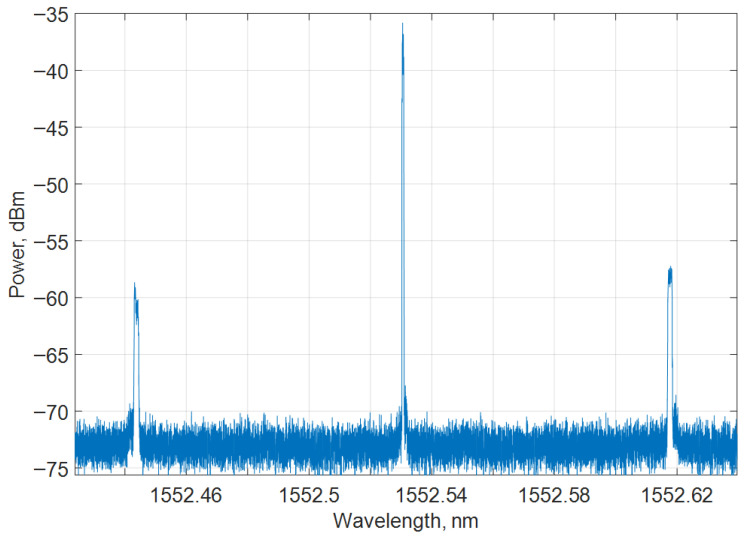
Experimentally measured spectrum of Brillouin scattering.

**Figure 10 sensors-25-05283-f010:**
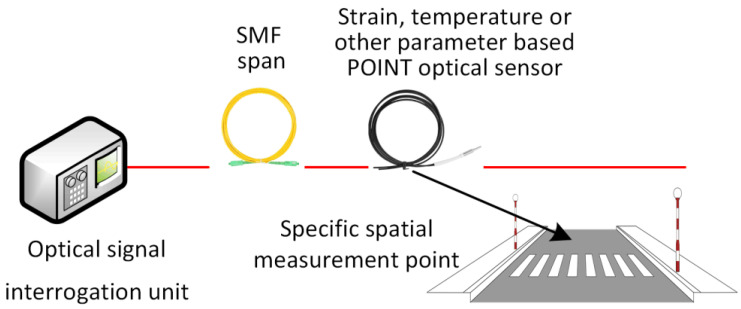
Single-point optical sensor-based solution within road architecture.

**Figure 11 sensors-25-05283-f011:**
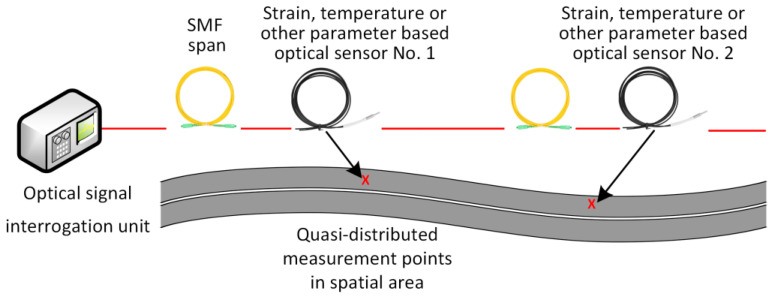
Quasi-distributed optical sensor-based solution within road architecture.

**Figure 12 sensors-25-05283-f012:**
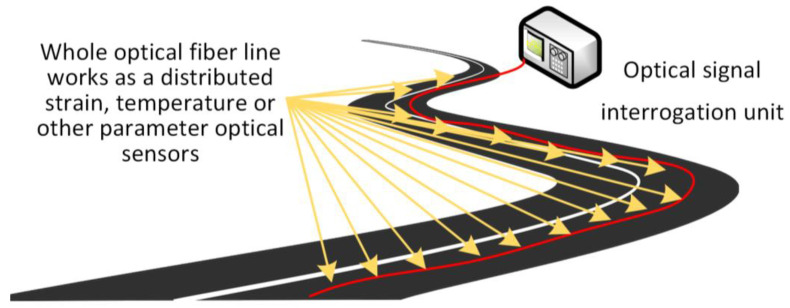
Distributed optical sensor-based solution within road architecture.

**Figure 13 sensors-25-05283-f013:**
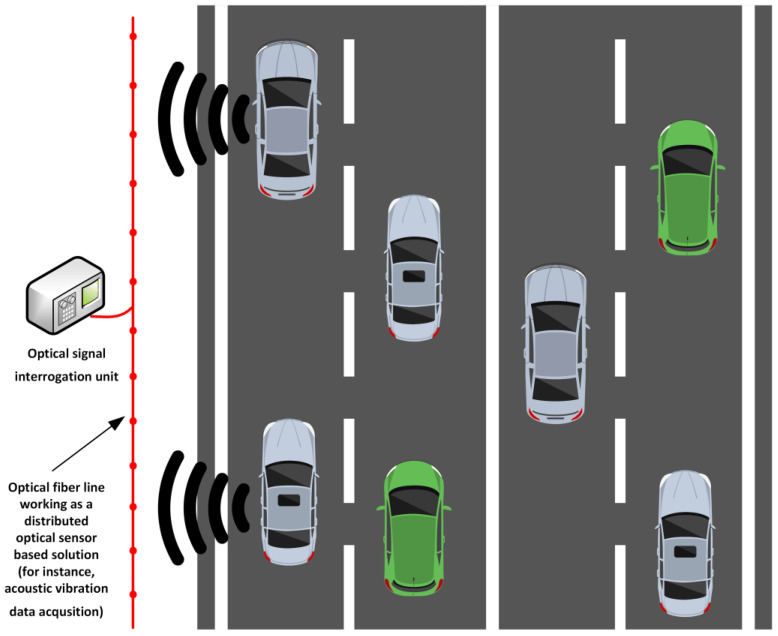
The concept for the distributed optical sensor realization in traffic volume monitoring.

**Figure 14 sensors-25-05283-f014:**
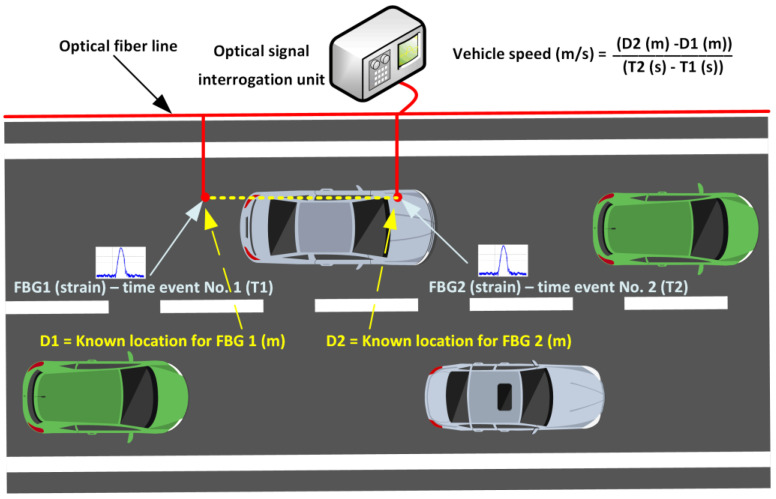
The concept of vehicle-induced strain detection and data correlation to the vehicle movement speed calculation.

**Figure 15 sensors-25-05283-f015:**
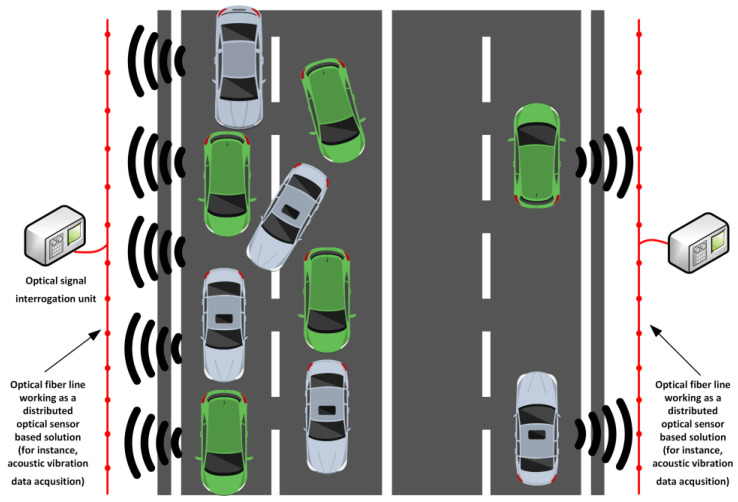
The concept of vehicle-induced strain detection and data correlation to traffic density monitoring.

**Figure 16 sensors-25-05283-f016:**
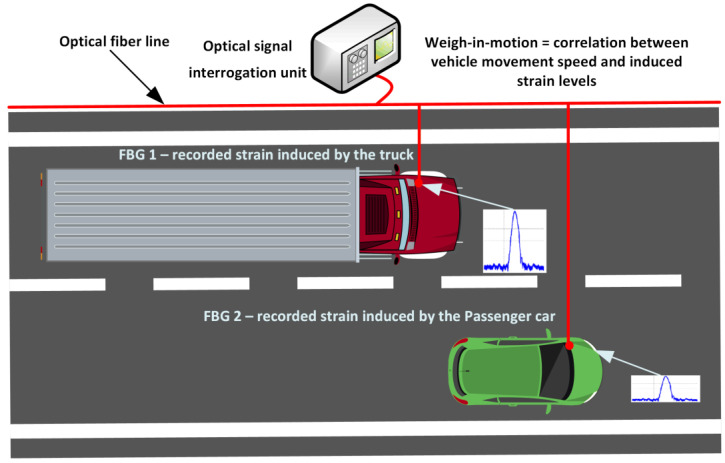
The concept of vehicle-induced strain detection and data correlation between vehicle movement speed and induced strain levels for weigh-in-motion applications.

**Table 1 sensors-25-05283-t001:** Comparison between topical optical sensor-based technologies and their technical capabilities [44,46,51,67,71,74,75,76,86,87,102,103,104,105,106,107,108,109,110,111,112,113,114,115,116,117,118,119,120,121,122,123,124,125,126,127,128,129,130,131,132,133,134,135,136,137,138,139].

Technology	Typical Sensitivity	Spatial Resolution	Monitoring Distance	Typical Application Type
Fiber Bragg Grating (FBG)	**Strain:** ∼1 pm/µε [51,105]	Point sensor: Typically, 1 mm to 1 cm [114,115,116]	From short (a few meters) up to 100 km: With the use of Wavelength Division Multiplexing (WDM) and appropriate amplification [87,117]	Localized strain, vibration, pressure, humidity, and temperature sensing: Commonly used in structural health monitoring, aerospace, and civil engineering applications [59,118]
	**Temperature**: Highly sensitive, typically 10 pm/°C [51,105]			
	**Vibration:** ~10–1000 pm/g [106,107]			
	**Pressure:** ~7–280 pm/kPa [108,109,110]			
	**Displacement and Tilt:** ~10–200 pm/° (tilt) [111,112]			
	**Humidity:** 0.1% RH [113]			
Rayleigh Scattering	**Strain**: ∼0.5–1 µε [86,119]	High spatial resolution: 1 mm to 1 cm [45,80] or a few meters [126]	Short to medium distances: Typically, up to 30 m; specialized systems can reach up to 2 km [86]	High-resolution distributed strain and temperature sensing: Suitable for applications requiring detailed spatial information over shorter distances [126,127]
	**Temperature**: ∼0.05–0.1 °C [86,119]			
	**Vibration:** pm/g not available, up to 350–500 Hz [120,121]			
	**Pressure:** 7 με/kPa [122]			
	**Displacement and Tilt:** ~0.05–1 mm [119,123]			
	**Humidity:** 1–2% RH [124,125]			
Brillouin Scattering	**Strain**: 10–50 µε [128,129]	Moderate spatial resolution: Typically, 0.2 m to 3 m [104,129,138]	Long distances: Ranges from 10 km up to 100 km, depending on system configuration [138,139]	Long-distance distributed strain and temperature sensing: suitable for monitoring large infrastructures like roads, pipelines, and bridges [102,103,104]
	**Temperature**: ∼0.28–1.3 °C [129,130,131]			
	**Vibration:** 55–60 Hz [46,132,133]			
	**Pressure:** 0.7–3.5 MHz/MPa [134]			
	**Displacement and Tilt:** 1 mm–15 cm over 1 m (displacement) [135,136]			
	**Humidity:** 0.4–1.65% RH [130,137]			
Raman Scattering	**Strain**: N/A	Spatial resolution: Approximately 1 m in longer distances [74,75] or 1cm with a 3 m distance [44]	Moderate distances: Typically, up to 10 km (ROFDR) [74] or extended to 20+ km (ROTDR) [75]	Distributed temperature sensing: Commonly used in fire detection and environmental monitoring [67,71,76]
	**Temperature**: ∼0.1 °C (ROFDR)–1.8 °C (ROTDR) [74,75]			
	**Vibration:** N/A			
	**Pressure:** N/A			
	**Displacement and Tilt:** N/A			
	**Humidity:** N/A			

**Table 2 sensors-25-05283-t002:** Comparative costs per sensing system and point for traditional and fiber-optic road monitoring technologies [34,36,37,59,61,66,71,75,92,93,110,145,146,148,149,150,151,152,153,154,155].

Method	Sensor Type/Technology	Measured Parameters	Cost Basis	Approx. Cost Per Sensing Point (€) *	Notes
Traditional Methods	Manual inspection	Cracks, surface defects, joints, drainage issues	Service (~200 €/km)	~2–4	Assumes ~50–100 inspected points per km. Cost per point estimated by dividing service cost by number of inspection points.
	Imaging (digital camera, laser profiler)	Surface roughness, rutting, cracks	Service (~110 €/km); system >€3000–5000	~1–3	Assumes ~50–100 processed points per km. Excludes capital equipment amortization.
	Falling Weight Deflectometer	Deflections	Geophones >€50 units; system >€50,000	>50	Cost per point = cost per sensor. Excluding the controller cost.
	Weigh-in-motion (WIM)	Weight, traffic flow, vehicle geometry and motion, temperature	Sensor €3500–40,000 (sensor cost per lane), system >€20,000	>3500	Cost per point = cost per lane. Excluding full-scale trailer and FWD costs.
	Inductive loop sensor system	Traffic flow, vehicle counts, speed	Sensor >€3000 per loop; system> 10,000	>3000	Cost per point = cost per loop. Excluding the controller cost.
Fiber-Optic Technologies	Quasi-distributed FBG sensors	Strain, temperature, pressure, displacement	FBG > €50 each; interrogator > €5000	>50	Cost per sensing point is the cost of an individual FBG, excluding the interrogator cost.
	Distributed Raman OTDR	Temperature	Sensing fiber: €0.3–few €/m; interrogator >€100,000	~0.3–few	Assumes 1 mm spatial resolution; cost per point = fiber price per meter. Excluding the interrogator cost.
	Distributed Brillouin OTDR	Strain, temperature	Similar to Raman	Similar to Raman	Assumes 1 m spatial resolution. Excluding the interrogator cost.
	Distributed Rayleigh OTDR	Strain, temperature	Similar to Raman	100 times smaller than Raman	Assumes 1 cm spatial resolution. Excluding the interrogator cost.

* Costs are indicative and may vary depending on supplier, configuration, quantity, and installation requirements. Estimates for “cost per sensing point” are based on typical commercial configurations and standard spatial resolutions.

## Data Availability

The data used to support the findings of this study are available from the first author upon request.

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
