# Peer review of "Comprehensive Analysis of FBG and Distributed Rayleigh, Brillouin, and Raman Optical Sensor-Based Solutions for Road Infrastructure Monitoring Applications"

_sensors, 2025, doi:10.3390/s25175283_

Round 1

Reviewer 1 Report

Comments and Suggestions for Authors

The publication is very interesting and presents a wide range of solutions for monitoring road structures. These range from vision-based solutions to numerous solutions using fiber optics.

While reading this article, I was missing two key things:

1. References to other (non-optical) road monitoring methods are missing.

2. There is no reference to sensors built with birefringent fiber optics.

In the reviewer's opinion, these two pieces of information should be added to the article to enhance its value.

Author Response

Thank You for the Review Report. Please see answers to comments in the appendix. 

Reviewer 2 Report

Comments and Suggestions for Authors

This paper presents a comprehensive review of fiber-optic sensing technologies—FBG, Raman, Rayleigh, and Brillouin systems—for road-infrastructure structural-health monitoring (SHM). It systematically compares traditional inspection methods (visual, FWD, GPR, IRT, etc.) with emerging optical-fiber approaches, analyzes each sensing technology’s working principles, sensitivity, spatial resolution, and achievable range, and discusses three deployment architectures: point, quasi-distributed, and fully distributed. Real-world road-monitoring applications (traffic volume, vehicle speed, density, weigh-in-motion) are illustrated, and technical trade-offs are summarized in tabular form. The manuscript is well-structured, the literature coverage is broad, and the conclusions are appropriately cautious. Several points need clarification or strengthening before publication.

  1. Provide a concise comparative table summarizing the cost per sensing point for each fiber-optic technology versus traditional methods.
  2. Adding a dedicated subsection on calibration and long-term drift issues for embedded fiber-optic sensors in asphalt concrete, including temperature–strain cross-sensitivity mitigation strategies. And the progress of temperature sensing with iron-ceramic enhanced fiber bragg grating sensors can be referred here, for examples, citing (1) “Khan, R. Y. M., Ullah, R. ., & Faisal, M. (2024). High-Temperature Sensing with Iron-Ceramic Enhanced Fiber Bragg Grating Sensors: Encapsulation Strategies and Concentration Dependencies. Journal of Optics and Photonics Research. https://doi.org/10.47852/bonviewJOPR42023202” (2) about DFB Temperature Sensing Based on Swept Laser Demodulation Technique: citing “Liu, Jia, Zhang, Bai, Liu, Yiwei, Femtosecond Laser Inscribed Multiwavelength Cascaded FBG Array Using Point-By-Point Method for Temperature Sensing Based on Swept Laser Demodulation Technique, International Journal of Optics, 2025, 3826450, 8 pages, 2025. https://doi.org/10.1155/ijo/3826450”. Moreover in the Introduction, please integrate recent advances on enhanced FBG technique in other area such as reliability analysis of residential buildings by citing: (3) “Zabihollah, A., & Shi, Y. (2025). Reliability Analysis of Residential Buildings Under Hurricane Using Embedded FBG Sensors: Remaining Useful Lifetime Analysis. Journal of Optics and Photonics Research. https://doi.org/10.47852/bonviewJOPR52024397”
  3. The authors can include at least one quantitative case study that reports measurement accuracy validated against a reference instrument (e.g., strain gauge or thermocouple) under controlled loading and temperature cycles.
  4. Why the spatial resolution of BOTDA is limited to ~1 m in field tests (Section 2.4), yet Rayleigh-OFDR claims millimetre resolution (Section 2.3)? Discuss whether this is a fundamental limitation or an instrumentation trade-off.
  5. State explicitly whether the raw datasets underlying Figures 9–11 can be made available in a public repository or upon reasonable request, in line with MDPI’s open-data policy.
  6. Ensure the acronyms FWD (Falling Weight Deflectometer) and TSD (Traffic Speed Deflectometer) are defined at first use in the main text, not only in the abstract.
  7. Expand the “Discussion” to outline quantitative performance targets (e.g., strain resolution <10 µÉ› over 10 km range) that would be required for network-level SHM adoption, and identify technological gaps to reach those targets.

Author Response

Thank You for the Review Report. Please see answers to comments and questions in the appendix. 

Reviewer 3 Report

Comments and Suggestions for Authors

The paper presents an analysis of single-point and distributed fiber-optic sensor applications for road infrastructure monitoring, as well as information on traditional monitoring methods with their advantages and disadvantages. Single-point FBG sensors, distributed Rayleigh, Raman and Brillouin scattering based sensors are analyzed, along with their unique characteristics, limitations, and applications.

The paper is well-structured and is beneficial to researchers in acquiring necessary information about the current state of road infrastructure monitoring using fiber-optic sensors. It provides a variety of references for further investigation. I recommend the paper "Comprehensive analysis of the FBG and distributed Rayleigh, Brillouin, and Raman optical sensor-based solutions for road infrastructure monitoring applications" to be published in the Sensors journal.

Comments:

  • Some equations seem to have different sizes and problems with index presentation (eq. 12-15).

Author Response

Thank You for the Review Report. Please see the answers to the comments in the appendix. 

Round 2

Reviewer 2 Report

Comments and Suggestions for Authors

Publish